# Outcome of Ivermectin in Cancer Treatment: An Experience in Loja-Ecuador

**Yuliana Jiménez-Gaona** [1,2,*] **, Oscar Vivanco-Galván** [3] **, Gonzalo Morales-Larreategui** [1] **,
Andrea Cabrera-Bejarano** [1] **and Vasudevan Lakshminarayanan** [4,5]

[1] Departamento de Química y Ciencias Exactas, Universidad Técnica Particular de Loja (UTPL), San Cayetano Alto S/N, Loja PC1101608, Ecuador
[2] Instituto de Instrumentación Para la Imagen Molecular I3M, Universitat Politécnica de Valencia, E-46022 Valencia, Spain
[3] Departamento de Ciencias Biológicas y Agropecuarias, Universidad Técnica Particular de Loja (UTPL), San Cayetano Alto S/N, Loja PC1101608, Ecuador
[4] Theoretical and Experimental Epistemology Lab, School of Optometry and Vision Science, University of Waterloo, Waterloo, ON N2L3G1, Canada
[5] Department of Systems Design Engineering, Physics, and Electrical and Computer Engineering, University of Waterloo, Waterloo, ON N2L3G1, Canada
* Correspondence: ydjimenez@utpl.edu.ec

**Abstract:** (1) Background: Cancer is one of the leading causes of death worldwide, and trends in cancer incidence and mortality are increasing over last years in Loja-Ecuador. Cancer treatment is expensive because of social and economic issues which force the patients to look for other alternatives. One such alternative treatment is ivermectin-based antiparasitic, which is commonly used in treating cattle. This paper analyzed ivermectin use as cancer treatment in the rural area of the Loja province and the medical opinion regarding the use of ivermectin in humans. (2) Methods: The study used a mixed methodology using different sampling techniques such as observation, surveys, and interviews. (3) Results: The main findings show that 19% of the participants diagnosed with cancer take medicines based on ivermectin as alternative therapy to the cancer control and treatment without leaving treatment such as chemotherapy, radiotherapy, or immunotherapy, while 81% use it to treat other diseases. (4) Conclusions: Finally, we identify that the interviewed not only use IVM as anticancer treatment, but it is also used as a treatment against other diseases. Although the participants' opinions indicate that they feel improvements in their health after the third dose, the specialist considers that there is no authorization to prescribe these alternative treatments. In addition, they confirmed that currently, there is no scientific knowledge about the application of these treatments in humans and they do not recommend their application. Thus, the anticancer mechanism of ivermectin remains to be further investigated; therefore, we consider that it is important to continue with this research by proposing a new stage to evaluate and determine the pharmacological action of this type of drug through an in vitro study in different cultures of cancer cells.

**Keywords:** antiparasitic; cancer; ivermectin; treatment

## 1. Introduction

Cancer cells can result from the unregulated division of different types of cells in the human body and produce changes in the genes that control the functioning of cells, especially in growth and division [1]. These additional cells can be divided without interruption, forming masses called tumors [2]. It is found that the number of new cancer cases is increasing, thus constituting one of the leading causes of death worldwide [3]. It is also the second cause of death in Latin America and the third leading cause of death in Ecuador [4,5]. According to the American Cancer Society (ACS), cancer will kill 5.5 million women per year worldwide by 2030. This alarming number is attributed to the increase

in known cancer risk factors linked to physical inactivity, genetics, poor nutrition, obesity and reproductive causes, which could increase breast cancer risks [6]. A study by the International Agency for Research on Cancer (IARC) of the World Health Organization (WHO) showed that prostate, breast, cervical, lung, colorectal, and stomach cancers make up 63% of cases and 49% of deaths in Latin America [7].

Globocan 2018 [8] estimated a new number of cancer cases worldwide in 2018 for South America [9]. Likewise, at the national level, the number of reported cancer cases in Loja is alarming when compared to cities with large populations such as Quito, Cuenca, El Oro, Guayaquil and Manabí (Figure 1). For example, the reference [10] reports incidence and mortality rates of various cancer types; in the case of men, the prostate cancer incidence ranks second nationally in Loja, after the Quito city. In addition, SOLCA (Sociedad de Lucha Contra el Cáncer) reports that the most frequent and worrying cancer cases in Loja province are cervical, stomach, thyroid, breast, prostate, cancer, leukemias and lymphomas, followed by skin cancer.

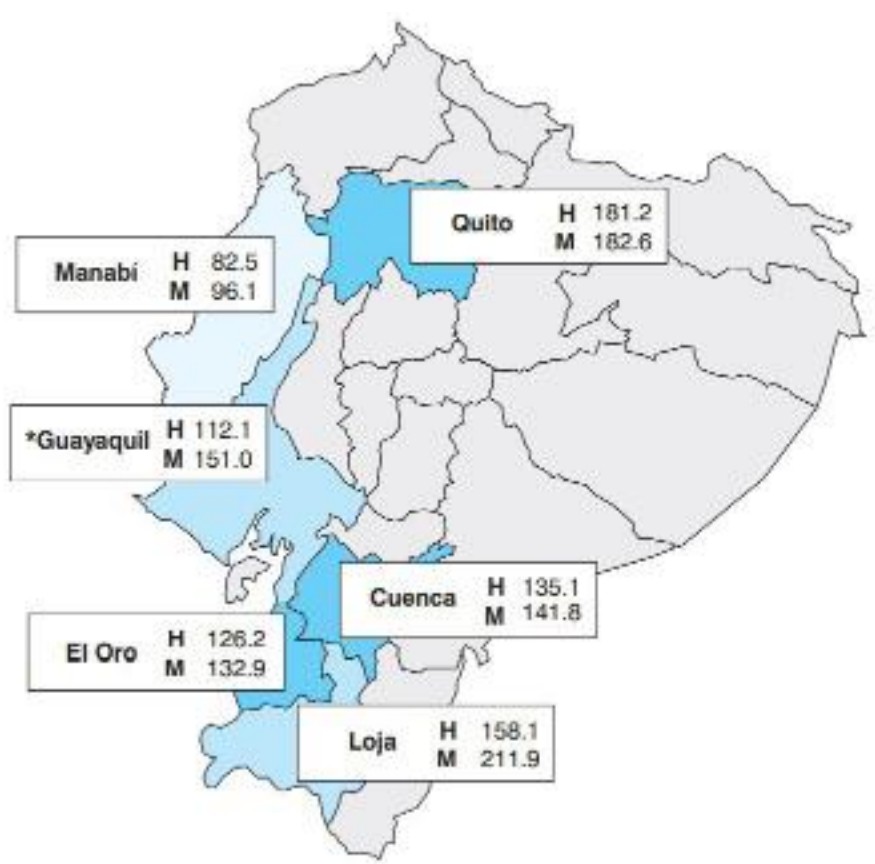

**Figure 1.** Standardized incidence rates in cancer registry in Ecuador. Taken from Epidemiology of Cancer in Quito 2006–2010 (2014). Capital letters correspond to H (Male) and M (Female) respectively. *: in Guayaquil, only the crude death rate is available.

On the other hand, another study [10] analyzed data over a seven-year (1997–2003) period on the incidence of cancer in Loja; there were 3566 cases of cancer of which 1479 were male and 2105 were female. Thus, Loja occupies the first place of cancer cases in ovarian, uterus, leukemia, gallbladder and bile ducts, colon–rectum and stomach in women. It is also the second in the incidence of stomach and leukemia cancer [11] in Ecuadorian males. Despite these numbers, access to specialized health centers (SHC) for cancer identification and treatment in Ecuador is limited even though reports show indicators of improvement in survival rates.

Consequently, there has been an enormous increase in cancer research, involving extensive biological and pharmacological studies aimed at developing new chemother-

apeutic drugs that selectively eliminate harmful cells [12–17]. However, self-medication has become a public health problem due to the side effects of the medications [18]. Hence, self-medication is considered inappropriate when the frequency of a prescription, duration or drug dose is higher than indicated, as well as the use of drugs with a high interaction risk (drug–drug or drug–disease), due to the use of inappropriate drugs [19].

For example, in Loja-Ecuador, there is an urgent need to find alternative short-term, low-cost cancer treatments that include the use of antiparasitic solutions that are ivermectin (IVM)-based. These medicines belong to the endectocides therapeutic group [20]. Other macrocyclic lactones are currently the most applied anthelmintics in the UK in the sheep industry [21] and in cattle.

Commonly, the injectable solution contains the IVM basis. The IVM is an old antiparasitic drug, which is considered to have antimitotic, anticancer potential and is an internal and external antiparasitic agent [21] derived from the avermectins that are a mixture of avermectin A1, A2, B1 and B2, a compound produced by the *Streptomyces avemitilis*, and has effective and prolonged control of gastrointestinal parasites in animals [22,23]. In addition, new IVM effects have recently been discovered as antiplasmodics, antiviral and mycobacterial and have been described as an antileukemic agent [24–27] and as an inhibitor of SARS-CoV19 replication [28,29]. IVM has been reported to inhibit the proliferation of several tumors' cells [30], and this also suggests that it can suppress almost completely the growth of various human cancers, including breast [30], colon, ovarian and melanoma [16,17,31]. Table 1 shows referenced studies about the IVM application as an anticancer drug to several subtypes of cancers.

**Table 1.** Summary of the anticancer IVM studies.

| Reference | Type of Study | Description | Objective | Subtype of Cancer |
|---|---|---|---|---|
| Draganov et al. [32] | In Vivo | The IVM effects were study using a 4T1 mouse model of TNBC. The results indicate that IVM has dual immunomodulatory and ICD-inducing effects in breast cancer | To evaluate animal line cells | Breast cancer |
| Dou et al. [29] | In Vivo and In Vitro | The authors report a role for IVM-induced autophagy in breast cancer cells. The results provide a molecular basis for the use of IVM to inhibit the proliferation of breast cancer cells. | To evaluate cellular PAK 1 line | Breast cancer |
| Zhang et al. [33] | In Vivo and In Vitro | The molecular mechanism and effects of IVM alone and its combination with cisplatin on growth and survival were examined. Results indicate that IVM significantly augmented the inhibitory effect of cisplatin on ovarian cancer cells in a dose-dependent manner. | To evaluate cultured ovarian cancer cells and a xenograft mouse model, focusing on Akt/mTOR signaling | Ovarian cancer |
| Jiang et al. [34] | In Vivo and In Vitro | The effects of IVM on cancer cells lines which are resistant to the chemotherapeutin drugs vincristine and adriamycin were investigate in vitro. Flow cytometry, immunohistochemistry, and immunofluorescence were used to investigate the reversal effect of IVM in vivo. Results indicated that IVM at its very low dose drastically reversed the resistance of the tumor cells to the chemotherapeutic drugs both in vitro and in vivo. | To evaluate two (HCT-8) colorectal cancer cells and (MCF-7) breast cancer cells), one hematologic tumor (K562) and two xenograft mice models | Colorectal, breast cancer and one hematologic tumor cell line |

**Table 1.** *Cont.*

| Reference | Type of Study | Description | Objective | Subtype of Cancer |
|---|---|---|---|---|
| Zhou et al. [35] | In Vivo | The results demonstrated that ivermectin dose-dependently inhibited colorectal cancer SW480 and SW1116 cell growth, which was followed by promoting cell apoptosis and increasing Caspase-3/7 activity. | To evaluate the influence of ivermectin on CRC using CRC cell lines SW480 and SW1116 | Colorectal cancer |

These findings warrant further IVM investigation for therapeutic cancer treatment [36].

Hence, this research was initiated through a sampling phase, using surveys and interviews from medical specialists as well as people with cancer in certain rural areas of the Loja province, to address some key elements of the antiparasitic IVM use in this population. This study addresses the following aims: (i) determine the IVM use in Gonzanamá and Catamayo areas in Loja province; (ii) explore the opinion of medical professionals regarding the use of this product as an alternative to cancer treatment.

## 2. Materials and Methods

For this multiphase research, a mixed experimental design was applied integrating qualitative (observational and descriptive study) and quantitative data (surveys and interviews) from subjects who met certain inclusion criteria (people who have received IVM as treatment for some type of cancer, vulnerable people with diseases considered catastrophic as cancer) and exclusion criteria (ignorance of the drug, people who refuse to participate in the study, subjects under 18 years old, pregnant women and people with some degree of disability).

The execution of the project was carried out through three progressive phases (Figure 2), which we detail below:

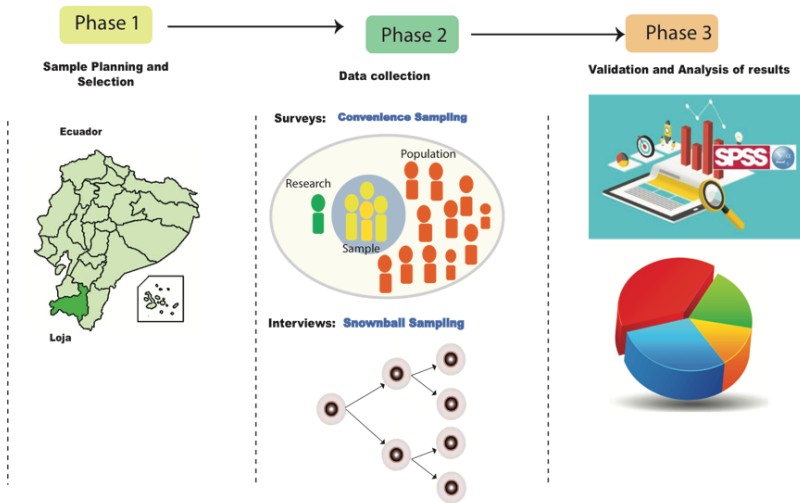

**Figure 2.** Implementation phases project. Phase 1: Sample Planning and Selection involves preparatory activities to define sampling techniques and delimit the area of study. Phase 2: Data Collection is used for the application of surveys and interviews. Phase 3: Validation and Analysis of results using the SPSS tool.

*2.1. Sample Planning and Selection*

Initial Requirements: In This Phase the Type of Sampling, Evaluation Metrics and the Instruments Preparation were Planned

Instruments: In this phase, the interview and survey were elaborated as well as the validation and reliability of the instruments to obtain information from the selected sample (cancer patients and doctors).

Sample: Data collection was carried out in two cantons of the Loja province: Catamayo and Gonzanamá (Changaimina, Sacapalca, Paja Blanca).

Universe: Loja is in the southern region of the Ecuador in Latin American. The planning zone 7 corresponds to: El Oro, Zamora Chinchipe, and Loja. Loja city occupies a territory of about 11,065.6 km$^2$, which is constituted by 16 cantons (Calvas, Catamayo, Célica, Chaguarpamba, Espíndola, Gonzanama, Loja, Macara, Olmedo, Paltas, Pindal, Puyango, Quilanga, Saraguro, Sozoranga, Zapotillo) with 448.966 inhabitants (Zonal Agenda 2013–2017).

*2.2. Data Collection*

2.2.1. Survey of Participants with Cancer

An ad hoc survey was used, with a non-probabilistic convenience sampling, with the aim of the interviewer effectively identifying people with cancer and progressively others with the same pathology.

Because of the large population, voluntary participants were selected according to the convenience, accessibility, and proximity to the researcher. Subjects with certain risk factors (lack of community cooperation to provide information about the use of these types of medicines) were eliminated from the study, thus ensuring a better outcome.

The first part of the survey focused on collecting anthropometric and social parameters from the sample; the second part collected information about the type of oncological diseases and other pathologies. The last part was aimed at collecting information about the usage and amount of IVM dosage, and lastly, we investigated the therapeutic response to treatment and side effects of self-medication.

A total of 48 patients aged between 18 and 94 years were surveyed over a three-month period from January–March 2020. The survey collected data on age, weight, gender, community, commercial name of the alternative medicine, dose (mL), frequency, approximate time of use, side effects, discontinuation time of chemotherapies or related medical therapies. The procedure is shown in Figure 3a,b.

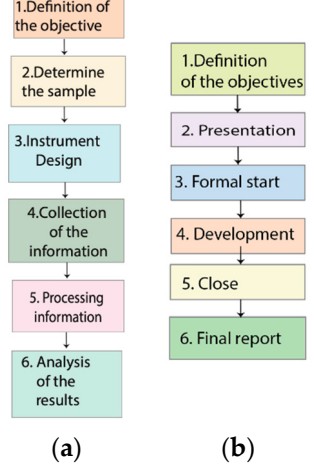

(**a**)          (**b**)

**Figure 3.** (**a**) Flowchart of the survey execution cycle. (**b**) Flowchart of the interview execution cycle.

2.2.2. Interview with Medical Oncologists

A non-probabilistic sampling type "snowball" was used for the interview. The instrument was applied to five medical oncologist's medical specialists from local institu-

tions such as Anti-cancer Society is Sociedad de Lucha Contra el Cáncer (SOLCA) (Quito, Ecuador) and the Hospital of Universidad Técnica Particular de Loja (H-UTPL). Their opinion about the use and the possible effects of the IVM application in humans was collected. Details are described in the results section (Table 1). The procedure used to collect information through interviews is shown in Figure 3b.

The interview was structured with a series of questions, where the first four questions evaluated the doctor's knowledge about local cancer conditions. The other three questions focused on perception of the use of IVM administration for cancer treatment. The final questions dealt with the possible solutions or controls of the IVM as an alternative cancer treatment.

### 2.3. Data Analysis and Interpretation

The descriptive statistics was used: measures of central tendency, measures of variability and graphs. All analysis was carried out through the SPSS v22 program. Qualitative variables were analyzed by frequencies/percentages, and quantitative variables (ordinal, nominal) were analyzed by measures of central tendency.

#### Confidentiality of Information

For information protection and security, personal data were anonymized and removed from the survey. Only the participating researchers had access to information that was completely anonymous and confidentially processed data coded questions. All data were stored in secure files accessed only by password. However, the interviews were not confidential since the study was designed to highlight the professional approach of the experts on this subject matter.

### 3. Results

The survey instrument consists of fifty different types of questions (open-ended, dichotomic, demographic and mixed) from where only six questions are based on Likert scale with five acceptable items. The reliability was validated using Cronbach's alpha score (0.76). A Shapiro–Wilk test was applied to evaluate the normality of data, obtaining a $p$-value > 0.05. The sample size was 48 persons; the demographic information is shown in Table 2.

**Table 2.** Demographic characteristics of respondents.

| Group | Sub-Group | Frequency (N) | Percentage (%) |
|---|---|---|---|
| Gender | Male | 20 | 41.67 |
| | Female | 26 | 54.17 |
| | Undefined | 2 | 4.16 |
| Age | 20–40 | 8 | 16.7 |
| | 41–60 | 27 | 56.3 |
| | 61–80 | 8 | 16.7 |
| | 81–100 | 5 | 10.3 |
| Rural area | Catamayo | 30 | 62.5 |
| | Changaimina | 1 | 2.08 |
| | Gonzanama | 3 | 6.26 |
| | Paja Blanca | 1 | 2.08 |
| | Sacapalca | 13 | 27.08 |

In the second part of the survey, information related to the cancer type, drugs and therapies used as treatment was collected. The results indicate that 18.75% of interviewees had cancer and took IVM as treatment. Among them, men exhibited prostate cancer (4.17%) and stomach cancer (6.25%). In women, 2.085% had breast cancer, 4.17% had cervical cancer and 2.085 had bonds cancer. The rest of the people surveyed correspond to 81.25%, who indicated other diseases such as diabetes, osteoporosis, gastritis, ovarian cysts, sinusitis, skin conditions, fatty liver, allergies, etc.

Similarly, 83.33% of interviewees confirmed the use of medicines based on IVM as a panacea for treating any illness; however, only 39.58% of all interviewees know that it is a veterinary antiparasitic. In addition, inferential statistical data (correlation and regression) were obtained with respect to the weight and quantity of the dose applied (Table 3). Table 4 indicates that the dose correlates significantly ($p$-value < 0.05) with the weight; however, it does not correlate with the age. Table 5 shows the ANOVA statistical analysis between the dose-dependent and the weight-constant variables, which indicates that there is a significant regression ($p$-value < 0.05). Likewise, the correlation between dose–weight (dose is the independent variable) indicates that correlation is significant at 0.05 with 0.95 of confidence in the result, as shown in Table 6.

**Table 3.** Correlations between age, weight, and dose.

|  |  | Age | Weight | Dose |
|---|---|---|---|---|
| | Pearson correlation | 1 | 0.260 | −0.004 |
| Age | Sig. (bilateral) | | 0.080 | 0.982 |
| | N | 48 | 48 | 32 |
| | Pearson correlation | 0.260 | 1 | 0.381 * |
| Weight | Sig. (bilateral) | 0.080 | | 0.031 |
| | N | 48 | 48 | 32 |
| | Pearson correlation | −0.004 | 0.381 * | 1 |
| Dose | Sig. (bilateral) | 0.982 | 0.031 | |
| | N | 32 | 32 | 32 |

*: The correlation is significant at the 0.05 level (two-tailed).

**Table 4.** Regression between weight and dose.

| Model | R | R Square | R Fitted Square | Standard Error |
|---|---|---|---|---|
| 1 | 0.381 [a] | 0.146 | 0.117 | 0.316 |

a: predictor variable.

**Table 5.** Statistical Analysis ANOVA [a].

| | Model | Sum of Squares | df | Root Mean Square | F | Sig. |
|---|---|---|---|---|---|---|
| | Regression | 0.509 | 1 | 0.509 | 5.109 | 0.031 [b] |
| 1 | Residue | 2.991 | 30 | 0.100 | | |
| | Total | 3.500 | 31 | | | |

a: dependent variable (dose); b: constant variable (weight).

**Table 6.** Coefficients and dependent variable: dose.

| Model | Non-Standardized Coefficients | | Standardized Coefficients | T | Sig. |
|---|---|---|---|---|---|
| | B | Standard Error | Beta | | |
| 1 (Constant) | 1.433 | 0.311 | | 4.605 | 0.000 |
| Weight | 0.010 | 0.004 | 0.381 | 2.260 | 0.031 |

Another part of the survey collected information about medical treatments used as alternative cancer therapy. It was determined that there are several types of medical treatments used to treat their disease. For example, 4.35% had surgical intervention, 13.04% had undergone chemotherapy and 6.52% had received radiation therapy. The remaining 76.09% indicated that they had used other alternative treatments, 28.18% indicated that they used natural medicine, and finally, 47.92% selected antiparasitic medicines based on IVM as a treatment option.

Likewise, the interviewees were asked about the frequency and dose application of the antiparasitic medicine. Thus, 43.75% specified that a dose was administered every three months, 35.42% specified that one dose was given once a month, 10.42% specified that one dose was given occasionally and rarely the 8.33%, while the 2.08% never applied the medication. About the amount applied, 12.50% indicated that it was generally 1–2 mL and 87.50% indicated that it was generally 3–5 mL (intramuscular) and claimed to have had positive effects after its application. It should be noted that only 8.33% had side effects such as diarrhea, skin blisters and pain from injection and burning sensation. The other cases reported not feeling any side effects.

Furthermore, 22.91 % of the sample said they had stopped medical treatments such as chemotherapy and radiotherapy over six months ago and had replaced them with IVM medicine. To all this, we can add a series of factors that the respondents consider important before the decision to use an antiparasitic drug as a medical therapy. These factors include the low cost of the drug, positive recommendations from neighbors, family members and friends who had indicated improvements in gastrointestinal discomfort, joint pains and even people diagnosed with cancer who felt improvements from the first application. This information was confirmed because the 81.25% of interviewees was satisfied with the IVM effect.

*Perception and Medical Opinion about the Possible Drug Effects on the Patient's Health*

A structured interview with open questions was conducted to obtain the medical specialist's opinion. The responses obtained were grouped into main ideas and presented in Appendix A. The category column indicates the fundamentals of the interview, and the subcategory column indicates the most frequently mentioned responses.

Thus, the medical specialists' interview results clearly indicated that there is no scientific certainty about the antiphrastic IVM effects in cancer patients. Likewise, oncologists confirm that scientific knowledge about the application of these drugs in humans is currently unknown, and therefore, they do not recommend their application. In addition, it is not within the authorized drugs for use in public health.

## 4. Discussion

There is currently inequality in medical and pharmaceutical access in rural areas of Loja province. Thus, not all people with cancer have access to different chemotherapy treatments, leading them to self-medicate and as well as search for alternative therapies.

From this study, information was obtained regarding the use of IVM as an alternative medicine not only against cancer but also against several diseases. We found evidence to assert that these people are self-medicating with veterinary antiparasitic solution as an alternative to other types of cancer treatment. This use has spread in the population since it is an affordable and accessible alternative to chemotherapy in people with certain types of neoplasia.

This application is not new, since there is evidence that IVM can serve as adjunct treatment for different diseases [37,38] and may even have a substantial value in the treatment of a variety of cancers, most of which affect the world's poor people [39]. For example, Liu et al. [40] show that an anthelmintic drug, IVM, is active against glioblastoma cells in vitro and in vivo, and targets angiogenesis. Dryinaev et al. [41] mention that IVM has a pronounced antitumor activity as well as the ability to enhance the antitumor action of Ehrlich carcinoma, melanoma B16 and lymphoid leukemia P388, including strain P388.

On the other hand, in 2016, scientists from Sichuan University, China and the Collaborative Innovation Center for Biotherapy revealed new IVM effects, such as high potential anticancer drug against colon, ovarian, melanoma, leukemia, renal cancer, angiogenesis and glioblastoma growth; the mechanism of action is unknown. Similarly, Hashimoto et al. [42] and Melotti et al. [43] point out that IVM would be able to inhibit tumor cell proliferation in humans, inducing autophagy by PAK1 for breast cancer regulation [28].

Other studies by Diazgranados-Sánchez et al. [38] have shown that the IVM is an excellent therapeutic alternative in neurocysticercosis management. These researchers tested IVM in patients who did not respond to conventional treatment with albendazole and/or praziquantel. They found no clinical deterioration, and the resonance imaging that was performed two and three months after treatment showed cerebral cysticercoses disappearance, decreasing the frequency of seizures in people with epilepsy despite non-modification of the regular anti-seizure treatment scheme. This study also described those patients diagnosed with epilepsy who received therapeutic IVM doses (10 mg) three times a week, with one dose/day orally, without modification of the medication that patients previously received except in cases of intolerance, such as drowsiness, skin rash or other affectation attributable to medications. The results show that during the IVM treatment time (between 12 and 24 months), patients demonstrated a decrease in the mean number of epileptic seizures.

However, as we mentioned above, we do not recommend these alternatives because they are not prescribed by doctors, and the mechanism of the antiparasitic solution based on IVM is unknown. In addition, the real impact in health and the side effects should be studied. Rather, we emphasize the importance of not abandoning the therapies prescribed by the doctor (surgery, chemotherapy, radiation therapy, immunotherapy, targeted therapy, among others) to all people who opt for adjunct cancer treatment.

Likewise, another important aspect to consider is the dosage that people receive. Thus, according to the study results, the dose is directly related to weight rather than age. Usually, the dose ranges from 1–2 mL to 3–5 mL administered intramuscularly once or twice a month. Some indicated that they felt positive effects after IVM application, and others indicated that they felt side effects such as diarrhea, vomiting, stomach pain, etc.

The IVM toxicity study [36] in humans gives us a better idea of side effects. This study indicates that a dose between 0.05 and 0.40 mg/kg does not cause unwanted effects and risk to human life; doses between 6.6 and 8.6 mg/kg are toxic, causing vomiting, blurred vision, mydriasis, ataxia, tremor, and coma, and finally, lethal doses are of 24 mg/kg.

IVM is a compound widely used in the world as a human antiparasitic with great efficacy and minimal or few unwanted effects. It has been widely used successfully as a treatment against COVID-19 [44–47]. However, care should be taken with IVM antiparasitic used in bovines, because other chemical compositions not only contain 1 mg of IVM but also contain 10 mg of Chlorsulon, in addition to other excipients (formal glycerol, Propilenglicol) that could be toxic to human health.

Our study had several strengths which are briefly described. (i) The findings offer new, potentially useful information for this patient population and for future molecular studies and determine some specific subtype of cancer that responds better to ivermectin treatment. (ii) The sample was collected by the snowball statistical method, which was a useful way to conduct our research about people with specific conditions who might otherwise be difficult to identify. This method is cheaper and generally available. In addition, it has some main limitations: (i) The first one is the small number of participants. Thus, (ii) it is necessary to include more rural areas whose people use ivermectin as antiparasitic or anticancer treatment.

In the future, we will develop a research plan that will be carried out as an intermediate phase through an in vitro molecular system using cancer cells, with a final phase in a biotherium. We are searching for more rigorous follow-up and validity to our findings and thus offer new opportunities to patients with few resources to improve their quality of life.

## 5. Conclusions

At the end of this research, it was possible to identify that the participants not only use ivermectin as an anticancer treatment, but it is also used as a treatment against other diseases.

The specialist considers that although there is no authorization to prescribe these alternative treatments by people who are not legally qualified, there is a shared responsibility with the patient, since these should not be consumed without medical authorization, nor

should patients be leaving aside a treatment or therapy prescribed by a specialist. In addition, they confirmed that currently, there is no scientific knowledge about the application of these treatments in humans, and they do not recommend their application.

Finally, we consider that it important to continue with this research by proposing a new stage to evaluate and determine the pharmacological action of this type of drug through an in vitro study in different cultures of cancer cells.

## 6. Future work

From the information obtained, we plan to establish the following phases: (1) Initiate new molecular research through an in vitro laboratory study, using prostate, ovarian and breast cancer cell lines. (2) Establish trials in a biotherium to experiment with biological reagents and protocols in animals to determine the effect of the antiparasitic drug IVM. (3) Determine if IVM has properties involved in regulating cell growth functions. (4) Generate knowledge about the possible effects of the active substance and disseminate them to the population through campaigns on the proper use of drugs.

**Author Contributions:** Conceptualization, Y.J.-G.; O.V.-G. and V.L.; methodology, Y.J.-G. and O.V.-G.; software, Y.J-G. and G.M.-L.; validation, G.M.-L. and A.C.-B.; formal analysis, G.M.-L.; investigation, Y.J.-G.; O.V.-G.; V.L. resources, Y.J.-G. and A.C.-B.; data curation, A.C.-B.; writing—original draft preparation, Y.J.-G.; O.V.-G.; V.L.; writing—review and editing, Y.J.-G.; O.V.-G.; V.L.; visualization, Y.J.-G.; O.V.-G.; V.L. supervision, V.L.; project administration, Y.J.-G.; All authors have read and agreed to the published version of the manuscript.

**Funding:** This research received no external funding.

**Institutional Review Board Statement:** The study was conducted in accordance with the Declaration of Helsinki and approved by the Institutional Review Board (Ethics Committee of Observational Research in Humans-CEISH) of Universidad Técnica Particular de Loja-Ecuador (protocol number UTPL-CEISH-2019-11, 29 November 2019) for studies involving humans.

**Informed Consent Statement:** Informed consent was obtained from all subjects involved in the study.

**Conflicts of Interest:** The authors declare no conflict of interest.

## Appendix A

**Table A1.** Supplementary data.

| Code | Category | Subcategory | Frequency |
|---|---|---|---|
| 1 | Key aspects of the medicine | IVM is a veterinary antiparasitic drug extrapolated to humans as an anticancer treatment. | 4 |
| | | Doesn't known | 2 |
| 2 | Diagnosis and treatment of cancer | Preventive examinations, biopsy, endoscopy, screening, mammography, colposcopy, prostate antigen, etc. | 5 |
| | | Treatment through radiotherapy, chemotherapy, surgery, etc. | |
| 3 | Harmful effects of the medicine | Affects other organs of the body, causes local pain, hematoma, sciatic nerve puncture, anaphylactic effect, allergic reaction, vomiting, etc. | 2 |
| | | It blocks the necrosis activities of cancer cells. | 2 |
| | | It has minimal side effects; it is neither neurotoxic nor hepatocytic. | 1 |
| 4 | Medical recommendations | They do not recommend the use of these drugs since there are no scientific studies on the subject. | 6 |
| | | Medical awareness is fundamental through campaigns, screening, preventive control, promoting attendance to specialized medical lefts such as SOLCA, IESS (Ecuadorian Institute of Social Security), public hospitals, etc. | 5 |
| | | The patient first needs to take the treatment suggested by his doctor. | 1 |
| 5 | Molecular research | It is important to perform an in vitro molecular study on the molecular composition of these drugs. | 4 |
| | | No, it is considered important. | 2 |

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
