# Peer review of "Outcome of Ivermectin in Cancer Treatment: An Experience in Loja-Ecuador"

_nursrep, doi:10.3390/nursrep13010030_

Round 1

Reviewer 1 Report

General Information about the manuscript 

The authors present a small study on the use of ivermectin to treat cancer in a population of individuals in Loja, Ecuador. These types of studies are an interesting introduction to how certain people will try various treatments for terminal diseases when contemporary medical treatments are not available or affordable. 

I agree that people should be made aware of these types of studies, but at the same time, I am concerned that these types of studies give those with ulterior motives additional information to incorrectly use and misconstrue endpoints that fit their desired narrative.

The presented research is interesting and should be reconsidered for publication after the authors address the following major concerns.  Please see below. 

Comments on the manuscript

I found this article interesting for several reasons. The collection of medical information from a population that might not trust “authority figures” is impressive. The SARS-CoV-2 pandemic demonstrated the lengths people would go (at least in the United States) not to follow sound scientific and medical advice and try possible treatment methods that have limited scientific scrutiny. The second paragraph of the Conclusion strongly recommends against the off-label use of ivermectin to treat cancer. The authors should consider strongly framing their study as a significant number of people use ivermectin to treat cancer, which is a concern. The flaw is the difficulty of identifying even 90% of the population using ivermectin to treat cancer. While reading the manuscript, it was hard for me to determine if the authors supported the use of ivermectin to treat cancer in an undocumented manner.

In the Abstract, the authors state, “Conclusions: according to the participant opinion’s who have used ivermectin-based medications felt improvement in their health after the third dose. However, the anticancer mechanism of ivermectin remains to be further investigated.” This passage made me wonder if it was possible that those using ivermectin might have had a parasitic infection that was alleviated through the ivermectin treatments/doses.

The authors need to highlight and definitively indicate which of the referenced studies were for in vitro and in vivo studies. Specifically, it would be helpful for the reader to know how ivermectin was evaluated and the target. For example, was the target a specific protein/enzyme or a cell line or in an animal model? This information will help the reader gauge the impact of the study.

If possible, it would be nice to know if there is a specific subtype of cancer that responds better to treatment with ivermectin. 

The authors state (line 216), “…side effects such as diarrhea, vomiting, stomach pain etc.” Are all the noted side effects gastrointestinal? Would it be possible to list all side effects and indicate the number of occurrences?

The authors state, “ These factors include the: low cost of the drug, positive recommendations from neighbors, family members and friends who had indicated improvements in gastrointestinal discomfort, joint pains and even people diagnosed with cancer who felt improvements from the first application.” Could this be the placebo effect for those with cancer who indicated they felt better after their first application of ivermectin?

The passage (lines 231-234) states, “Likewise, oncologists confirm that scientific knowledge about the application of these drugs in humans is currently unknown, and therefore do not recommend their application. Also because it is not within the authorized drugs use by public health.” is very telling. I completely understand and appreciate the need to study and report new therapeutics for all disease states. I often wonder about the validity of these types of studies as they can provide inconclusive data to support unsubstantiated claims.

Why are people using “veterinary anti-parasitic drugs”? When did this practice become widespread?

Methods Section

The authors state, “With regard to age, 6.5% are between 18-33 years, followed by 10% in the 34-43-year range, 15% between 54-63 years, 8% between 64-73 years, 1% between 74-83 years and the remaining 4% over 84 years.” This passage does not match previous statements. Earlier, the authors stated, “…exclusion criteria (ignorance of the drug, people who refuse to participate in the study, subjects over 65 years old, and under 18 years old, pregnant women and people with some degree of disability)…” (lines 99-100) and “A total of 48 patients aged between 18 and 35 years were surveyed” (lines 137-138). These discrepancies are a serious concern.

Figures and Tables

Figures 1 and 2 are rather small and hard to read. Larger images should improve the readability of the images within the figures.

The arrows in Figure 3 are not consistent. The arrows in the lower left corner of the cycles are stylistically different from the other corners.

Grammar and Style

The manuscript needs to be edited for clarity, syntax, and word choice.

Within the text, the authors use a dot (.) to denote the decimal point, yet within the tables, the authors use a comma (,) to denote the decimal point. Please be consistent.

When reporting p-values, the term p-value should be italicized in lowercase letters.

Author Response

Reviewer 1

Comment 1

I found this article interesting for several reasons. The collection of medical information from a population that might not trust “authority figures” is impressive. The SARS-CoV-2 pandemic demonstrated the lengths people would go (at least in the United States) not to follow sound scientific and medical advice and try possible treatment methods that have limited scientific scrutiny. The second paragraph of the Conclusion strongly recommends against the off-label use of ivermectin to treat cancer. The authors should consider strongly framing their study as a significant number of people use ivermectin to treat cancer, which is a concern. The flaw is the difficulty of identifying even 90% of the population using ivermectin to treat cancer. While reading the manuscript, it was hard for me to determine if the authors supported the use of ivermectin to treat cancer in an undocumented manner.

  • We appreciate your positive comments, and We consider ourselves to be impartial during this research, in accordance with what we mentioned in the second paragraph of the conclusions of the manuscript.

Comment 2

In the Abstract, the authors state, “Conclusions: according to the participant opinions who have used ivermectin-based medications felt improvement in their health after the third dose. However, the anticancer mechanism of ivermectin remains to be further investigated.” This passage made me wonder if it was possible that those using ivermectin might have had a parasitic infection that was alleviated through the ivermectin treatments/doses.

  • Perhaps we can include it in a future survey question to inquire about intestinal parasites and deworming in the interviewee. However, in this study we selected only people who use ivermectin as an injectable solution which is normally used in animals (not in humans), and not those who use ivermectin as a pill.

 Comment 3

The authors need to highlight and definitively indicate which of the referenced studies were for in vitro and in vivo studies. Specifically, it would be helpful for the reader to know how ivermectin was evaluated and the target. For example, was the target a specific protein/enzyme or a cell line or in an animal model? This information will help the reader gauge the impact of the study.

If possible, it would be nice to know if there is a specific subtype of cancer that responds better to treatment with ivermectin. 

  • Thank you for the suggestion. We added table 1. Summary of the anticancer IVM studies. in the introduction

Comment 4

The authors state (line 216), “…side effects such as diarrhea, vomiting, stomach pain etc.” Are all the noted side effects gastrointestinal? Would it be possible to list all side effects and indicate the number of occurrences?

Are all the noted side effects gastrointestinal? Would it be possible to list all side effects and indicate the number of occurrences?

  • The question in the survey was open-ended ¿Could you cite some (s) of the most important side effects? and not all effects reported were gastrointestinal, only four (8.70%) interviewees indicated had diarrhea, vomiting and stomachache from the total of 48 interviewees. The other cases reported not feeling any side effects.

 Comment 5

The authors state, “These factors include the: low cost of the drug, positive recommendations from neighbors, family members and friends who had indicated improvements in gastrointestinal discomfort, joint pains and even people diagnosed with cancer who felt improvements from the first application.” Could this be the placebo effect for those with cancer who indicated they felt better after their first application of ivermectin?

  • Sorry, but right now it is difficult for us to know about the placebo effect of ivermectin in cancer participants, because it was not part of our objective.

 Comment 6

The passage (lines 231-234) states, “Likewise, oncologists confirm that scientific knowledge about the application of these drugs in humans is currently unknown, and therefore do not recommend their application. Also, because it is not within the authorized drugs use by public health.” is very telling. I completely understand and appreciate the need to study and report new therapeutics for all disease states. I often wonder about the validity of these types of studies as they can provide inconclusive data to support unsubstantiated claims.

  • We appreciate your comment, but we believe this study reflects clearly evidence of the economic, social and health problems especially in Ecuador, where the inequality in medical and pharmaceutical access in rural areas is evident. Where, not all people with cancer have access to different chemotherapy treatments, leading them to self-medicate and use other economic alternative therapies as cancer treatment. Due these methods are not prescribed by doctors, because the mechanism of its impact is not known, we belief that it is a large field for further research, as well as pointed out in "future work".

Comment 7

Why are people using “veterinary anti-parasitic drugs”? When did this practice become widespread?

  • We appreciate your comment, but we consider that it is not part of our objectives to know when this veterinary practice started.

 Comment 8

Methods Section

The authors state, “With regard to age, 6.5% are between 18-33 years, followed by 10% in the 34-43-year range, 15% between 54-63 years, 8% between 64-73 years, 1% between 74-83 years and the remaining 4% over 84 years.” This passage does not match previous statements. Earlier, the authors stated, “…exclusion criteria (ignorance of the drug, people who refuse to participate in the study, subjects over 65 years old, and under 18 years old, pregnant women and people with some degree of disability)…” (lines 99-100) and “A total of 48 patients aged between 18 and 35 years were surveyed” (lines 137-138). These discrepancies are a serious concern.

  • Sorry, in the line 100 we corrected this typing error, removing this exclusion criteria ¨over 65 years old¨. In line 137 we corrected the value from 18 to 94 “A total of 48 patients aged between 18 and 94 years were surveyed”, according to the data collected.

 Comment 9

Figures and Tables

Figures 1 and 2 are rather small and hard to read. Larger images should improve the readability of the images within the figures.

Thank you, the figures were larger as you suggested.

 Comment 10

The arrows in Figure 3 are not consistent. The arrows in the lower left corner of the cycles are stylistically different from the other corners.

  • Thank you, the arrows in figure 3 a and b were modified.

    Comment 11

Grammar and Style

The manuscript needs to be edited for clarity, syntax, and word choice.

Within the text, the authors use a dot (.) to denote the decimal point, yet within the tables, the authors use a comma (,) to denote the decimal point. Please be consistent.

When reporting p-values, the term p-value should be italicized in lowercase letters.

  • The manuscript was edited for clarity and syntax, also we standardized in all documents the punctuation marks as you suggested.

Reviewer 2 Report

Submitted paper is a valuable contribution and relates to an alternative cancer treatment which is the use of the antiparasitic drug ivermectin.  Ivermectin is very common antiparasitic drug which is used very widely. Reports on the use of ivermectin as an anti-cancer drug bring new possibilities. Comparing these methods to another way to treatment cancer, it is cheaper and generally available methods. Authors analyzed the ivermectin as cancer treatment in area Loja-Ecuador, where cancer incidence are increasing. They used in methodology different methods such as observation, surveys, and interviews. The conclusions of the research results show that IVM is used in the treatment of cancer. However, it is not a method prescribed by doctors, because the mechanism of its impact is not known. This is a large field for further research, which has been  pointed out in the chapter "future work".

In the general the paper is ready for publication. The language used is of high quality. I think that the topic of research will be highly interest of the readers.

I have few comments about methodology, results and figures.

In “Materials and methods” in first paragraph are presented “exclusion criteria” among others “subjects over 65 years old, and under 18 years old”. In line 137 it is confirmed that you used patients between 18 and 35 years. 

But in results in paragraph 2 is written: “With regard to age, 6.5% are between 18-33 years, followed by 10% in the 34-43-year range, 15% between 54-63 years, 8% between 64-73 years, 1% between 74-83 years and the remaining 4% over 84 years.” So the study included patients over 65 years of age, which is contrary to the rejection criteria, adopted in the chapter on materials and methods. 

In addition, in the same paragraph in results: “With regard to age, 6.5% are between 18-33 years, followed by 10% in the 34-43-year range, 15% between 54-63 years, 8% between 64-73 years, 1% between 74-83 years and the remaining 4% over 84 years”, the percentage of participants in a given age range is incorrect because the sum is not 100%.

The comment about the figure 3. I would present the flowchart in a linear way, not circular form, because it presents a process where “analysis of the results/final report” is final step of interview/survey. It is my suggestion.

Author Response

Reviewer 2

The submitted paper is a valuable contribution and relates to an alternative cancer treatment which is the use of the antiparasitic drug ivermectin.  Ivermectin is a very common antiparasitic drug which is used very widely. Reports on the use of ivermectin as an anti-cancer drug bring new possibilities. Comparing these methods to another way to treat cancer, it is cheaper and generally available methods. Authors analyzed the ivermectin as cancer treatment in area Loja-Ecuador, where cancer incidence are increasing. They used different methods such as observation, surveys, and interviews. The conclusions of the research results show that IVM is used in the treatment of cancer. However, it is not a method prescribed by doctors, because the mechanism of its impact is not known. This is a large field for further research, which has been pointed out in the chapter "future work".

In the general the paper is ready for publication. The language used is of high quality. I think that the topic of research will be highly interest of the readers.

I have few comments about methodology, results and figures.

Comment 1.

In “Materials and methods” in first paragraph are presented “exclusion criteria” among others “subjects over 65 years old, and under 18 years old”. In line 137 it is confirmed that you used patients between 18 and 35 years. 

But in results in paragraph 2 is written: “With regard to age, 6.5% are between 18-33 years, followed by 10% in the 34-43-year range, 15% between 54-63 years, 8% between 64-73 years, 1% between 74-83 years and the remaining 4% over 84 years.” So the study included patients over 65 years of age, which is contrary to the rejection criteria, adopted in the chapter on materials and methods. 

  • Sorry, in the line 100 we corrected this typing error, removing the exclusion criteria ¨over 65 years old¨. In line 137 we corrected the value from 18 to 94 “A total of 48 patients aged between 18 and 94 years were surveyed”, according to the data collected.

Comment 2.

In addition, in the same paragraph in results: “With regard to age, 6.5% are between 18-33 years, followed by 10% in the 34-43-year range, 15% between 54-63 years, 8% between 64-73 years, 1% between 74-83 years and the remaining 4% over 84 years”, the percentage of participants in a given age range is incorrect because the sum is not 100%.

  • According to the demographic characteristics (table 2) this data was modified in relation to the ages in the results section.The sample size was 48 persons: 20 male (41.67%), 26 female (54.17%) and 2 (4.16) did not indicate their gender. Regarding age, 16.7% are between the range 20-40 years, followed by 56.3% in the 41-60 years, 16.7% between 61-80 years, 10.3% between 81-94 years.

Comment 3.

The comment about figure 3. I would present the flowchart in a linear way, not circular form, because it presents a process where “analysis of the results/final report” is the final step of interview/survey. It is my suggestion.

  • Thank you, figures 3 a and b were modified as you suggested.